# Dissecting peripheral protein-membrane interfaces

**Thibault Tubiana**[ID][1,2¤], **Ian Sillitoe**[3], **Christine Orengo**[3], **Nathalie Reuter**[ID][1,2]*

**1** Department of Chemistry, University of Bergen, Bergen, Norway, **2** Computational Biology Unit, University of Bergen, Bergen, Norway, **3** Department of Structural and Molecular Biology, University College London, London, United Kingdom

¤ Current address: Thibault Tubiana: Université Paris-Saclay, CEA, CNRS, Institute for Integrative Biology of the Cell (I2BC), 91198, Gif-sur-Yvette, France

* nathalie.reuter@uib.no

**Data Availability Statement:** All structures and code used in our study are available on github (https://github.com/reuter-group/tubiana_etal_2022) and archived on Zenodo (https://doi.org/10.5281/zenodo.6772161). The list of the structures in the S100 clusters and the list of Alphafold models

## Abstract

Peripheral membrane proteins (PMPs) include a wide variety of proteins that have in common to bind transiently to the chemically complex interfacial region of membranes through their interfacial binding site (IBS). In contrast to protein-protein or protein-DNA/RNA interfaces, peripheral protein-membrane interfaces are poorly characterized. We collected a dataset of PMP domains representative of the variety of PMP functions: membrane-targeting domains (Annexin, C1, C2, discoidin C2, PH, PX), enzymes (PLA, PLC/D) and lipid-transfer proteins (START). The dataset contains 1328 experimental structures and 1194 AphaFold models. We mapped the amino acid composition and structural patterns of the IBS of each protein in this dataset, and evaluated which were more likely to be found at the IBS compared to the rest of the domains' accessible surface. In agreement with earlier work we find that about two thirds of the PMPs in the dataset have protruding hydrophobes (Leu, Ile, Phe, Tyr, Trp and Met) at their IBS. The three aromatic amino acids Trp, Tyr and Phe are a hallmark of PMPs IBS regardless of whether they protrude on loops or not. This is also the case for lysines but not arginines suggesting that, unlike for Arg-rich membrane-active peptides, the less membrane-disruptive lysine is preferred in PMPs. Another striking observation was the over-representation of glycines at the IBS of PMPs compared to the rest of their surface, possibly procuring IBS loops a much-needed flexibility to insert in-between membrane lipids. The analysis of the 9 superfamilies revealed amino acid distribution patterns in agreement with their known functions and membrane-binding mechanisms. Besides revealing novel amino acids patterns at protein-membrane interfaces, our work contributes a new PMP dataset and an analysis pipeline that can be further built upon for future studies of PMPs properties, or for developing PMPs prediction tools using for example, machine learning approaches.

## Author summary

Peripheral membrane proteins (PMPs) are soluble proteins that bind transiently to the surface of cell membranes. Having the ability to exist in both a soluble and a membrane-

as well as their extracted segments are given in File S1. A complete dataset with all annotations at residue-level is also provided as in csv format (S2 File).

**Funding:** NR and TT acknowledge funding from the Research Council of Norway (Norges Forskningsråd, grants #251247 and #288008). The Research Council of Norway: https://www. forskningsradet.no/ The funders had no role in study design, data collection and analysis, decision to publish, or preparation of the manuscript.

**Competing interests:** The authors have declared that no competing interests exist.

bound form their membrane-binding region is constrained to retain a fine balance of polar and hydrophobic character, which makes it difficult to distinguish it from the rest of their surface. As a result peripheral membrane-binding sites are notoriously difficult to predict.

We collected and curated a dataset containing 2500 structures of PMPs and compared their membrane-binding sites to the rest of their solvent-accessible surfaces, in order to reveal features of PMPs'membrane-binding sites. We find that, among positively charged amino acids, lysines are significantly more present than arginines. Protruding hydrophobes are a landmark of the interfacial binding sites of ca. 2/3 of peripheral membrane binding proteins, indicating that a majority of PMPs takes advantage of the hydrophobic effect while a non-negligible minority (1/3) most likely relies on electrostatics interactions or other mechanisms. The IBS of peripheral membrane proteins contain significantly more glycines than the rest of their surface. These findings and the collected dataset will be useful for the development of prediction models for membrane-binding sites of PMPs.

## Introduction

The surface of cellular and organelle membranes is the site of intense activity. Besides membrane-embedded proteins, myriad soluble proteins bind to the surface of cellular membranes with exquisite resolutions in space and time. Such proteins, referred to as peripheral membrane proteins (PMPs) play key roles in crucial processes including signaling cascades and lipid metabolism. Peripheral membrane protein is a term that includes a wide variety of proteins including membrane-targeting domains such as C1, C2, FYVE, PH, PX, ENTH and GLA [1–3], enzymes involved in lipid metabolism such as phospholipases [4, 5], membrane remodeling machines such as BAR domains [6] or ESCRTIII [7], and lipid-transfer proteins [8] to name a few.

PMPs have in common that they exist in a soluble and membrane-bound form, and bind transiently to the chemically complex interfacial regions of membranes. Because PMPs are soluble proteins [9], they have historically been more amenable to structural studies than transmembrane proteins, at least in their soluble form. However, and because they bind only transiently to the surface of membranes, the structure of their membrane-bound forms remains elusive for structural biology techniques. It results that peripheral protein-membrane interfaces are poorly characterized and remain high-hanging fruits as drug targets [9–11], in contrast to protein-protein [12, 13] or protein-DNA/RNA interfaces [14].

In this manuscript we refer to the membrane binding sites of peripheral proteins as interfacial binding sites (IBS). Traditionally IBSs have been described as being composed of a mixture of hydrophobic and basic amino acids [15]. The basic amino acids lysines and arginines, often organised as patches [16], contribute to non-specific electrostatics with the negatively charged membrane surface and may help the protein to reach the membrane in a binding-competent orientation [17]. In that orientation, the hydrophobic side chains can anchor into the membrane and intercalate between the lipid tails. According to the Wimley-White hydrophobicity scale for proteins at membrane interfaces, the most relevant hydrophobic amino acids are the three aromatics (tryptophan, phenylalanine and tyrosine), followed by leucine, isoleucine, cysteine and methionine. A more nuanced view of the peripheral binding model has emerged with the increasing number of experimental and computational approaches of individual PMPs, notably from molecular simulations [18, 19], revealing a variety of contributions from

specific amino acids types to the protein-membrane affinity. For example, the ratio between basic and hydrophobic amino acids varies between PMPs, from strong unspecific electrostatics to proteins with barely any basic cluster [20–22]. We have shown that lysine side chains might engage in hydrophobic contacts [22, 23] and that the role of aromatic amino acids appears to be more elaborate than a mere hydrophobic anchor. The side chains of tyrosines and tryptophanes are instead able to take advantage of a wider palette of chemical interactions and insertion depths than first thought [24–27].

Despite these advances, we still lack a general description of peripheral protein membrane interfaces and how IBS surfaces differ from soluble protein surfaces. The ability of PMPs to remain soluble limits the number of hydrophobic amino acids exposed at their IBS. For that reason their IBS might have amino acid compositions that do not depart significantly from water-exposed protein surfaces. In the absence of easily distinguishable amino acids patterns, structural patterns might help discriminating PMPs membrane binding sites from other surfaces. In particular, terms like "hydrophobic spikes" [28, 29] or "protruding loops" [30] have been used. Inspired by this we formulated in Fuglebakk et al. [1] a mathematical model to describe these structures that we coined *hydrophobic protrusions*. Briefly hydrophobic protrusions are defined from a protein structure as the hydrophobic amino acids whose Cβ atoms are a vertex of the convex hull calculated from that structure. Analysing 1012 protein structures belonging to 326 protein families, we could show that our model discriminates strongly between surfaces of membrane-binding and non-membrane binding proteins and that protruding hydrophobes were over-represented in about 2/3 of the PMPs in our dataset. We interpreted this result as reliable evidence of the importance of the hydrophobic effect in peripheral protein-membrane affinity for a large majority of the analysed PMPs.

In Fuglebakk et al. [1] we analysed surfaces of whole structures instead of focusing on the IBS, owing to the limited availability of experimental determination and annotation of interfacial membrane-binding sites. Here we circumvent that limitation by selecting a dataset of protein domains for which the membrane-binding region is reported for several domains in the superfamily and for which there exists sufficient structural data to derive statistics from the dataset. We effectively assume that the membrane-binding region within a superfamily consists of the same elements of the fold for all domains in that superfamily, a common assumption in functional annotation of protein domains. Our dataset counts 1328 protein domains belonging to 9 protein superfamilies that represent the diversity of PMP functions: membrane targeting domains (PH, PKCa-C2, Factor V Discoidin C2, PKCd-C1, PX), enzymes (phospholipase C/D, phospholipase A), lipid transfer proteins (START), and annexins. In addition we extend this dataset with 1194 additional structures from the recently released AlphaFold Protein Structure Database [31, 32]. Representative structures are shown on Fig 1 where the membrane-binding region of the proteins is shown in orange.

To analyse these two datasets, we take advantage of the hydrophobic protrusion model but also extend it to consider amino acids around protrusions. While our earlier work focused on *hydrophobic* protrusions, we here extend the analysis to a wider range of amino acids type to account for other forces involved in protein-membrane binding. To our knowledge this is the first study investigating amino acid composition and structural patterns of a large curated dataset of peripheral membrane-binding sites, and to which extent it is distinguishable from the rest of the domains' accessible surface.

## Results

We collected 1328 structures belonging to 9 superfamilies. In each superfamily we defined the membrane binding site as described in the Methods section, and sorted the amino acids into

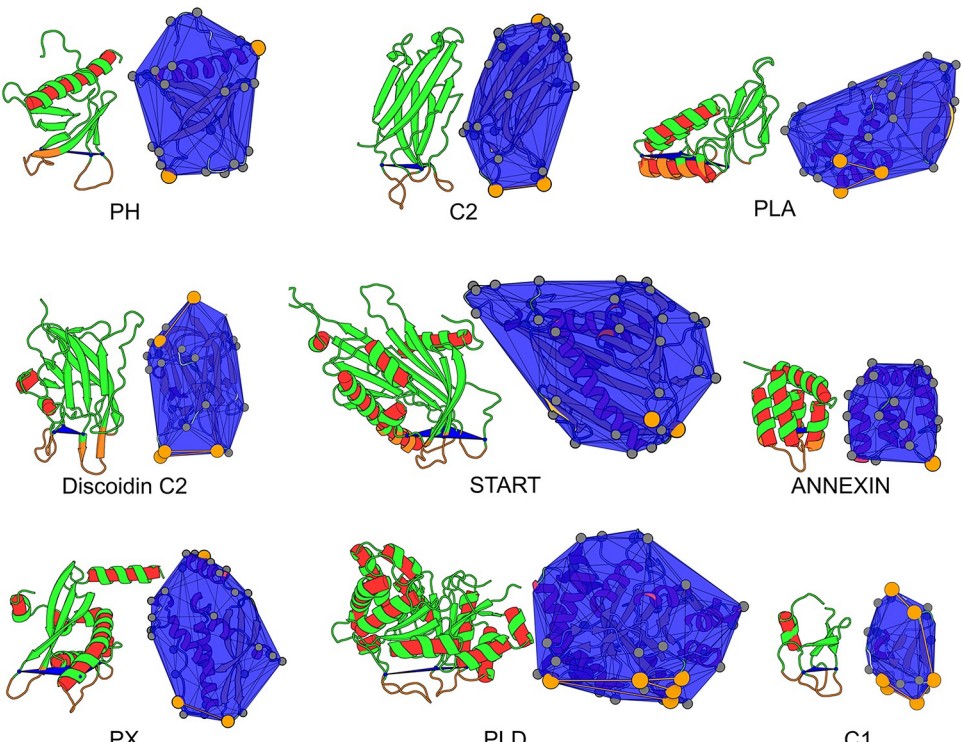

**Fig 1. Definition of the IBS, representative structure and hydrophobic protrusions for each superfamily.** (Left) Each representative structure is shown with cartoons, the IBS region is colored in orange. The plane defined by the three *reference* amino acids (cf. section on Data collection and processing in Material and methods) and delimiting IBS from non-IBS regions is drawn in blue. (Right) The convexhull of each domain is shown in blue, protrusions are marked by a gray sphere and hydrophobic protrusions orange spheres. Convexhull edges between hydrophobic protrusions are shown in orange. Orientation of each domain on left and right image may differ to increase visibility. Images were generated with the opensource version of PyMoL 2.5.0 [33] and PePr2vis (https://reuter-group.github.io/peprmint/pepr2vis) through Mol* [34].

the IBS and nonIBS datasets. Moreover for each structure we calculated the convex hull and identified protruding amino acids. Each amino acid in the IBS and nonIBS datasets is annotated with several features: whether it is a protrusion or not, or belonging to the neighborhood of a protrusion, the secondary structure it belongs to, its solvent accessible surface. The IBS dataset counts 27012 amino acids and the nonIBS datasets counts 156998 amino acids.

## 1. Amino acid distributions in the datasets of exposed IBS and nonIBS amino acids

A simple analysis of the amino acid composition of the IBS dataset (Fig 2A and 2B) shows very little difference to the nonIBS dataset (Fig 2C and 2D) but a few trends appear in terms of amino acid physicochemical properties (Fig 2A and 2C). The IBS dataset has more aromatics, less polar residues (Ser, Asn, Gln, Thr, His), and a bit more non-polar (Val, Ala, Gly, Pro) than the nonIBS datasets. There are nearly equal amounts of positive (Lys, Arg), negative (Asp, Glu) and hydrophobic (Leu, Ile, Cys, Met) amino acids in both datasets. We observe additional trends looking into individual amino acids (Fig 2B and 2D): there are slightly less Arg and Glu, and more Gly in the IBS than nonIBS datasets but the significance of this is difficult to assess.

This analysis is fairly coarse and overall shows an amino acid composition for the IBS dataset that is not very different from the nonIBS, and from what we know about protein exposed

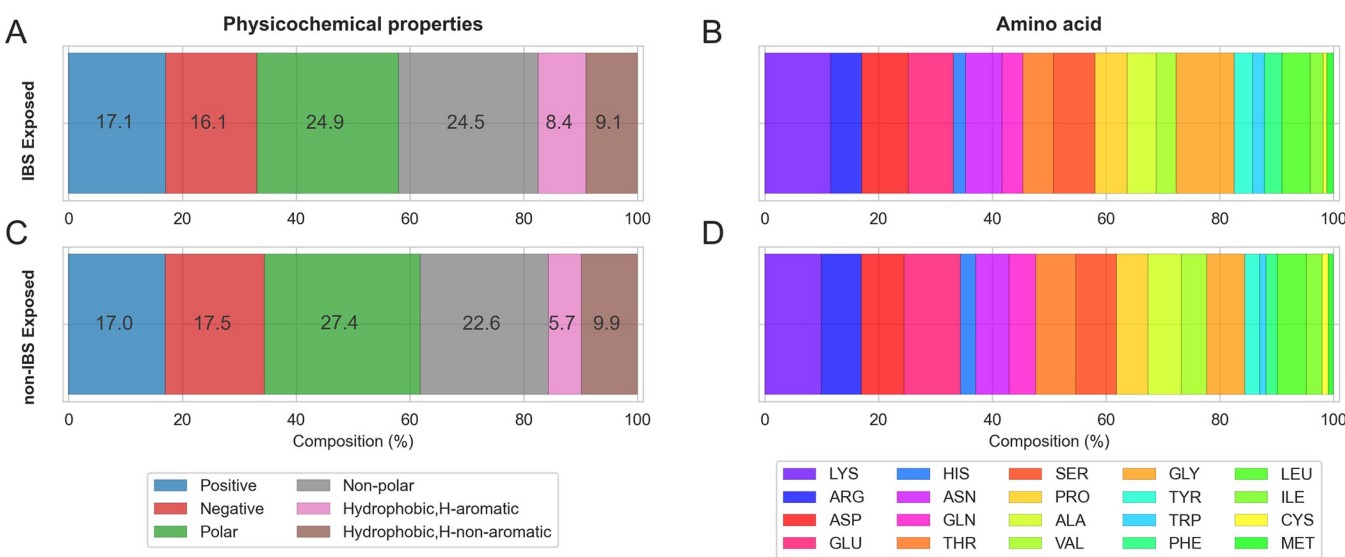

**Fig 2. Exposed amino acids in the IBS and nonIBS datasets.** Composition is calculated across all superfamilies and grouped by amino acid properties (plots A,C) (positive, negative, polar, nonpolar) and the 20 amino acids types (plots B,D) for amino acids belonging to the exposed IBS surface (A,B) and exposed nonIBS surface (C,D).

surfaces. It is not entirely surprising as PMPs need to remain soluble and therefore retain a reasonable balance between polar and hydrophobic amino acids, even at their IBS, not unlike the rest of PMPs surfaces. We thus expect differences between IBS and the reste of the PMP surfaces to be subtle.

This analysis reveals the need for a more detailed analysis to dissect the properties of PMPs IBS. In what follows, we will use our mathematical model of *hydrophobic protrusions* defined in Ref.[1] and extended as described in the Methods section.

## 2. Protrusions and hydrophobic protrusions in the IBS and nonIBS datasets

We first calculated the number of protrusions and hydrophobic protrusions per structure in the dataset (Fig 3). Fig 3A shows an average number of 27.0 ± 5.7 protrusions per protein on the whole surface and Fig 3B shows 7.1 ± 2.5 protrusions on average in the IBS region. This means that protrusions are present on every protein structure, but they are naturally less numerous at the IBS, which is only a fraction of the convex hull vertices of the protein (24% ± 8%). Looking only at hydrophobic protrusions, the difference between the whole surface and the IBS is less pronounced with an average number of 1.1 ± 1.2 at the IBS vs. 2.9 ± 1.9 on the whole surface. A total of 518 structures, 39% of the dataset, have no hydrophobic protrusions at their IBS, in agreement to what we reported earlier using a different dataset [1]. This is also visible on Fig 3C, where we report the number of structures with a given ratio of hydrophobic protrusions to the total number of protrusions. When looking at ratio of hydrophobic protrusions above 20%, the number of structures satisfying this condition at the IBS is higher than those satisfying it on the rest of their surface. This trend is confirmed by a Mann-Whitney test with an alternative hypothesis that the distribution of hydrophobic protrusions at the IBS is greater than in the nonIBS dataset. The P-value ($2.97.10^{-7}$) indicates that the two populations are indeed different and that the distribution for the IBS is greater than for the nonIBS dataset. This confirms that hydrophobic protrusions are more relatively present at the IBS than on the rest of the protein surface [1]. In what follows we analyse two groups separately, the structures with hydrophobic protrusions (810 structures) and those without protrusions (518 structures).

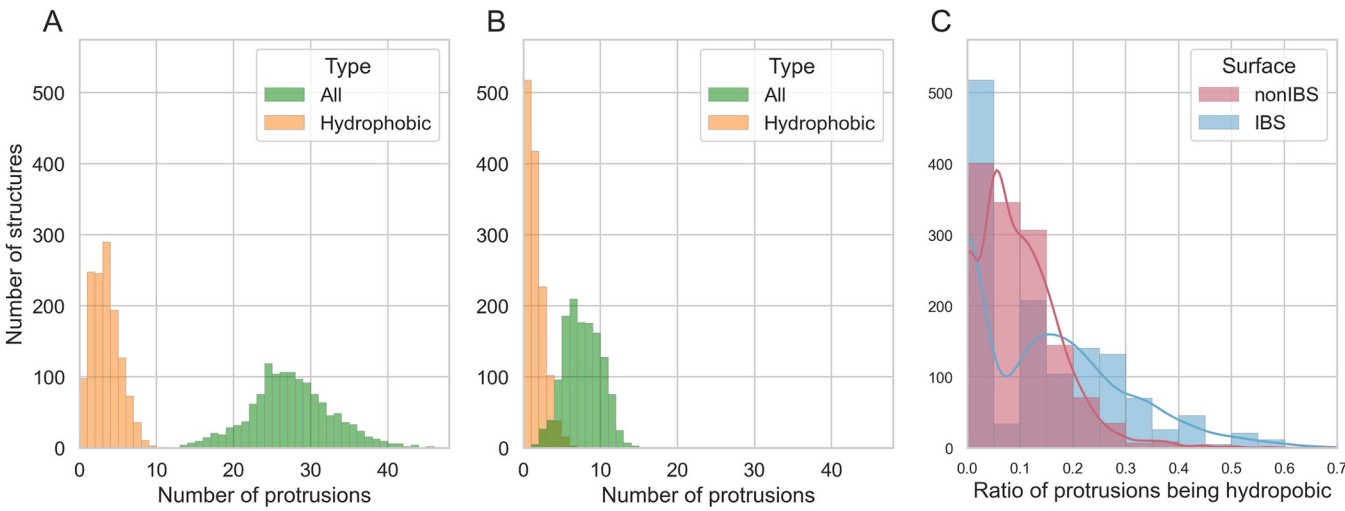

**Fig 3. Number and types of protrusions.** (A,B) Distribution of the protrusions in dataset structures, all protrusion types (green) and hydrophobic protrusions only (orange), calculated from all convex hull vertices (A) or only from vertices at the IBS (B). (C) Number of protein structures with a given ratio of hydrophobic vs non-hydrophobic protrusions on the whole surface (red) and at the IBS (blue). The lines represent the corresponding Gaussian density estimations.

## 3. Domains with hydrophobic protrusions at the IBS

Fig 4A shows the secondary structure elements (SSE) to which hydrophobic protrusions belong. Not surprisingly, the majority of protrusions are located on loops (including coil, bend and turn) at the IBS (71.7%) and on the rest of the protein surfaces (non-IBS, 66.1%). According to the corresponding odd ratio (OR) value (Fig 4A, bottom plot), protruding loops are more likely to be observed at the IBS. Interestingly we also observe that 25.2% of the protrusions are on α-helices, showing that the protrusion model can capture helical protruding segments, which is important since amphipatic helices are common at protein-membrane

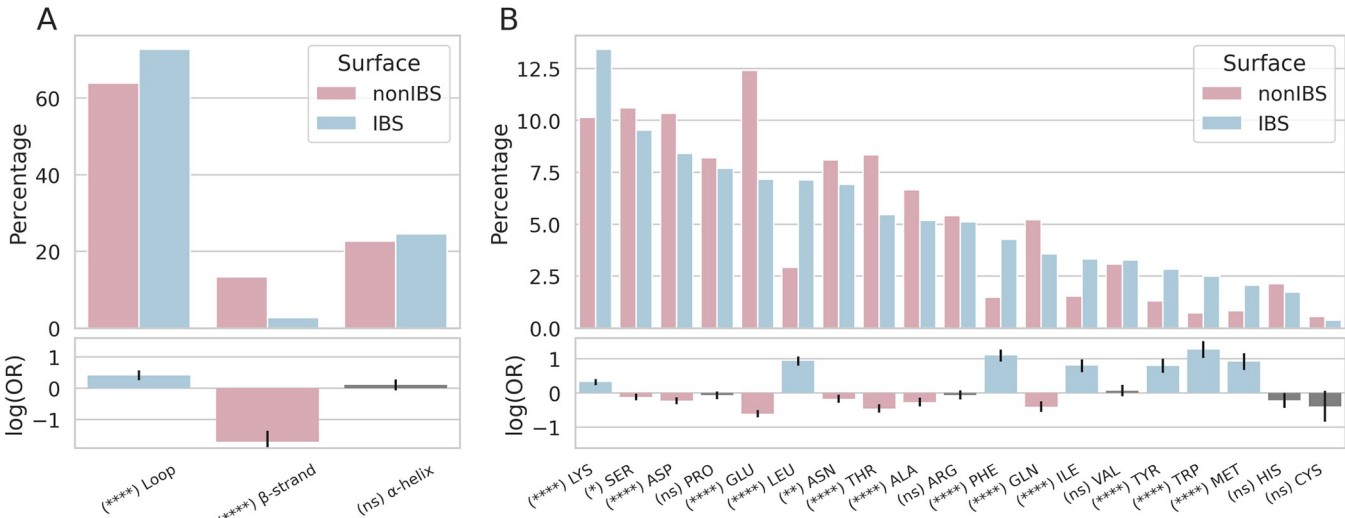

**Fig 4. Amino acid composition of protrusions in IBS and nonIBS subsets, for domains containing at least one hydrophobic protrusion at their IBS.** Secondary structure elements of hydrophobic protrusions (A) and amino acid composition for all types of protrusions (B) at the IBS (blue) and on the rest of the domain (nonIBS, pink) with respective odds ratio values. Grey bars for log(OR) indicate a P-value under 0.05 and, therefore, non-statistical differences between the IBS and nonIBS datasets. The analyses are restricted to the proteins with at least one hydrophobic protrusion at the IBS.

interfaces. There is no significant difference though between the IBS and the nonIBS datasets, indicating that helical protrusions are not more common at the IBS than on the rest of the protein surface. Unsurprisingly, the least likely SSE for a protrusion is β-strand with only 12.2% of protrusions outside of the IBS, and even less at the IBS (3.0%), with a negative log(OR) value.

Fig 4B shows the amino acid composition of protrusions at the IBS and in the nonIBS dataset for domains that have at least one hydrophobic protrusion. Fig 5 shows the amino acid composition of the neighborhood of these protrusions.

Hydrophobic protrusions consist mostly of large hydrophobes: Leu, Phe, Ile, Tyr, Trp, Met (Fig 4B). The respective odd ratio values are positive showing that these six amino acids are more likely to be found on protrusions at the IBS than on the non-binding surface of PMPs. These six hydrophobes represent together about 21% of all protrusions, with Leu present on ca 7% of the IBS protrusions. Lysine is the most common amino acid protruding at the IBS as it represents over 13% of all protruding amino acids. Moreover lysines on protrusions are more likely at the IBS than on the other regions of PMP surfaces (positive significant log(OR)). This is not the case for arginines that are equally present on IBS and nonIBS protrusions. Among the amino acids commonly protruding we also observe Ser, Asp, Pro and Glu but they are not more likely to be found at the IBS. Actually the log(OR) values for Ser, Asp and Glu are negative. It is also worth noting that Ser and Asp are in general highly frequent in loops (S2 Fig) and since protrusions are primarily located on loops (Fig 4A) it is not surprising to find these amino acids on protrusions.

The frequency of glycine in the neighborhood of hydrophobic protrusions at the IBS is strikingly high, and more likely than in the neighborhood of hydrophobic protrusions on the rest of the protein surface. The percentage of lysines is also high as we observed for protrusions

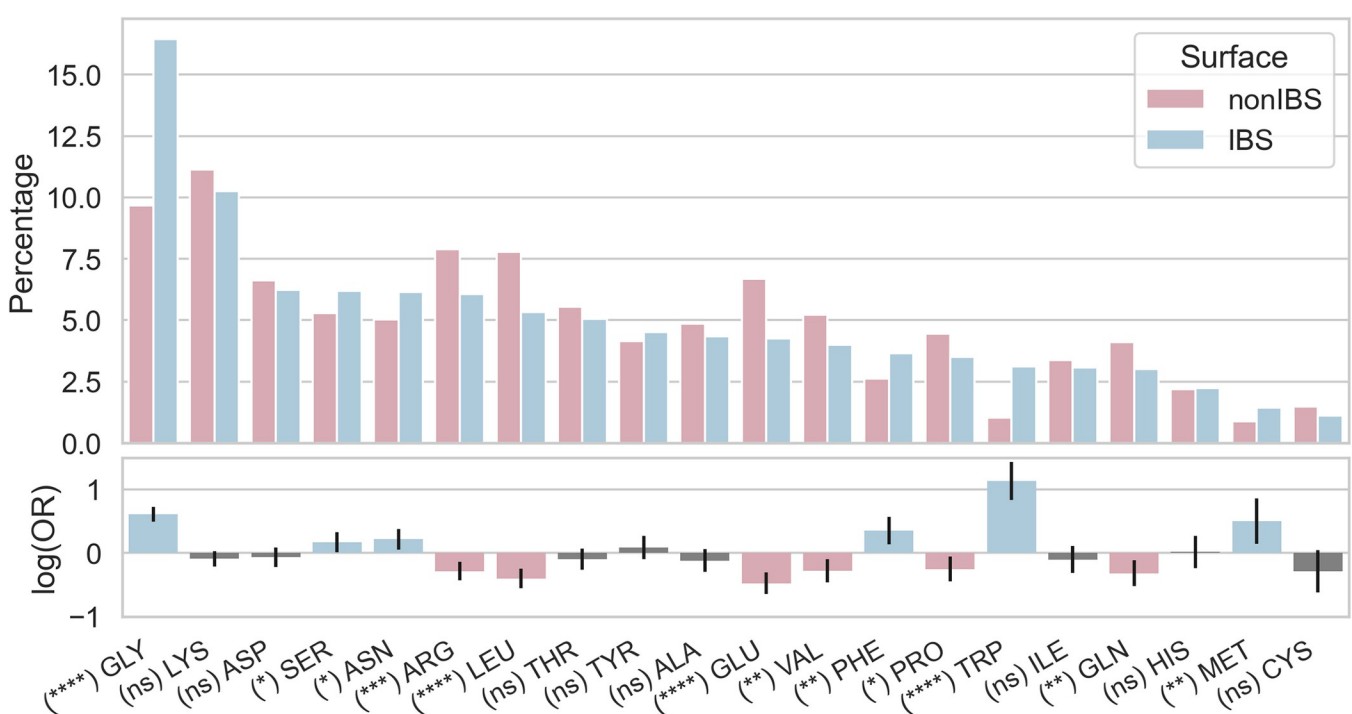

**Fig 5. Amino acid composition of the exposed neighbourhood of hydrophobic protrusions in IBS and nonIBS datasets.** Blue bars are for IBS amino acids and pink for non-IBS amino acids. Protrusions themselves are excluded. Grey bars for log(OR) indicate a P-value under 0.05 and therefore non-statistical differences between IBS and nonIBS datasets. The composition in secondary structures is reported in Supporting Information (S3A Fig).

themselves (Fig 4B), and they are more frequent than arginines. Finally, the percentage of aromatic residues is low at the IBS or outside (max 4%). Trp and Phe, and not Tyr, are more likely to be present around IBS protrusions than protrusions on the rest of the protein.

## 4. Proteins without hydrophobic protrusions at the IBS

As stated above the overall ratio of peripheral proteins in our dataset without hydrophobic protrusions at the IBS is about 39%, consistent with earlier results using a different dataset [1]. The proteins without hydrophobic protrusions at their IBS have less protrusions in their IBS in general, so the absence of hydrophobic protrusions is probably just the consequence of the structure of these proteins at the IBS (ie few protrusions). This is shown by the data plotted on Fig 6A which compares the frequency of proteins with given numbers of IBS protrusions, for two subsets: those having at least one hydrophobic protrusion in the IBS (yellow bars) have a histogram mode equal to 9, while those without hydrophobic protrusions in the IBS (blue bars) have a histogram mode of 5 and a narrower gaussian-like distribution. A Mann-Withney test yields a P-value of $6.88.10^{-44}$ showing that the two distributions are significantly different.

We find proteins without IBS hydrophobic protrusions in every of the nine superfamilies that we investigated (Fig 6B): the ratio is above 50% for C2DIS and START, but only about 30% for PH, ANNEXIN, C2, PLD, C1 and very few members of the PLA and PX superfamilies have no hydrophobic protrusions.

These proteins still contain protrusions at their IBS, and interestingly these consist mostly of Asp, Lys, Glu and Ser (Fig 7A). Arg are much less frequent than Lys, but as Lys and Asp, they are more likely to be found at IBS protrusions, indicating a pattern characteristic of this subset. This is the case for Ser and Pro as well. The environment of these non-hydrophobic protrusions resembles that of the hydrophobic protrusions (Fig 5) with a prevalence of Gly and Lys, and fewer Arg than Lys. We do not observe more hydrophobes in the neighborhood of non-hydrophobic protrusions than we observed in the neighborhood of hydrophobic protrusions (Fig 5). In other words, there is no compensation in the neighborhood for the lack of hydrophobic protrusions. Aromatic amino acids show a low percentage (2–4% each) but their

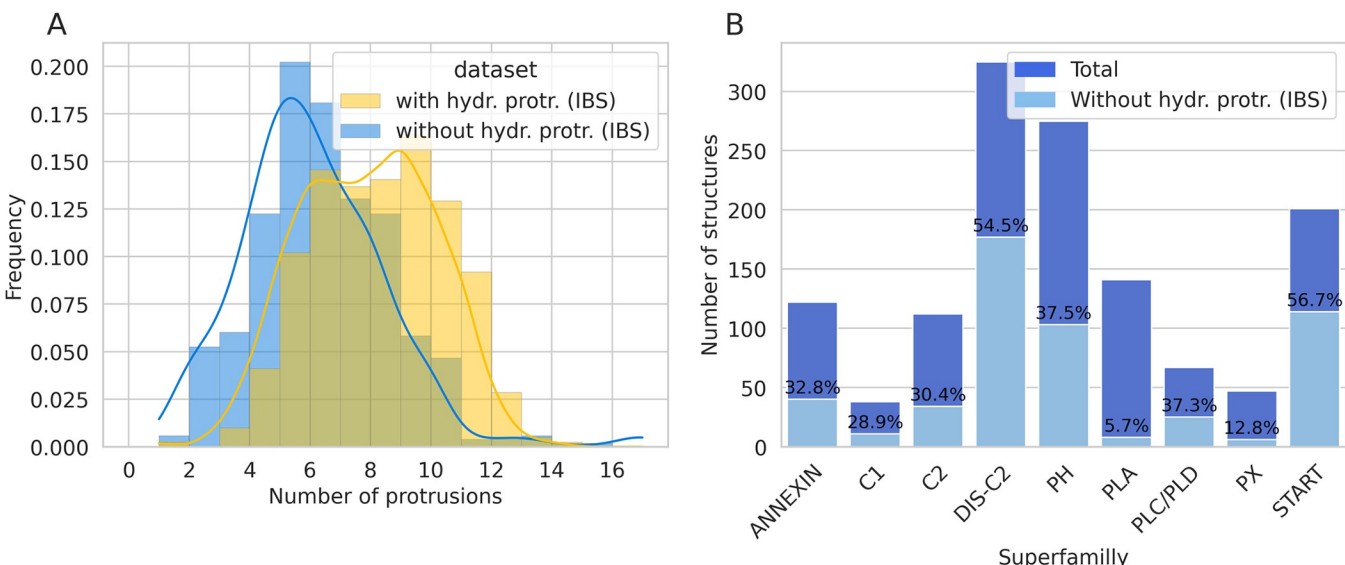

**Fig 6. Number of protrusions per protein and percentage of protein without hydrophobic protrusions.** (A) Distribution of domains according to their number of protrusions at the IBS, calculated for proteins with at least one hydrophobic protrusion in the IBS (blue) and proteins without hydrophobic protrusions at the IBS (yellow). The lines represent the respective gaussian density estimation. (B) Total number of structures in each of the nine superfamilies with their respective percentage of structures without hydrophobic protrusions in the IBS.

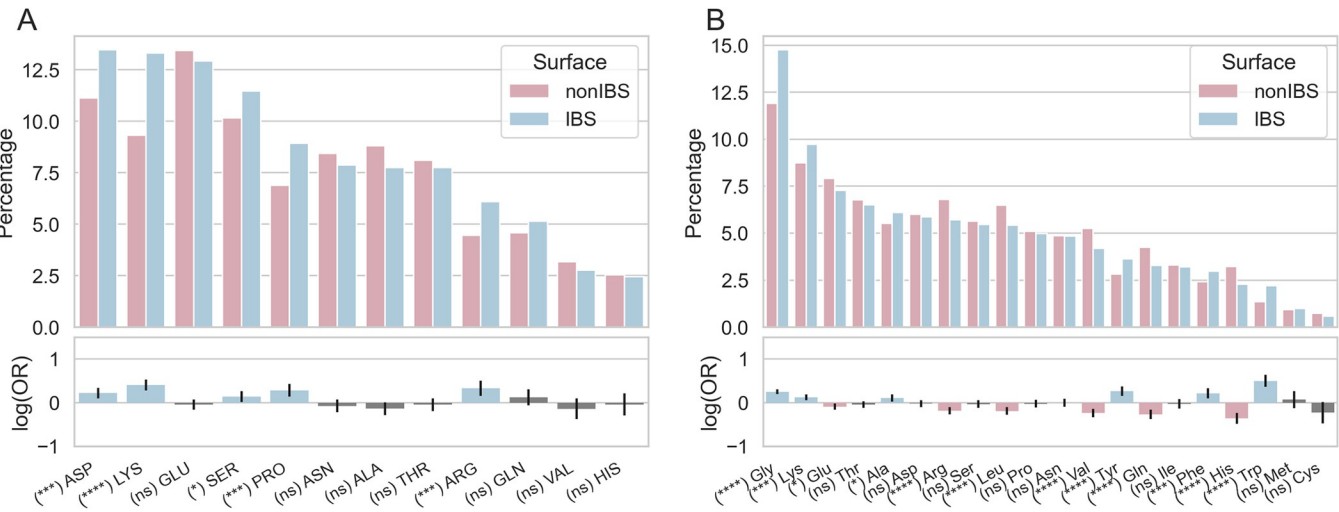

**Fig 7. Amino acids composing protrusions and their neighborhood for proteins without hydrophobic protrusions at their IBS.** Amino acid composition of (A) IBS (blue) and nonIBS (pink) protrusions, and (B) their neighborhood. Grey bars for log(OR) indicate a P-value under 0.05 and, therefore, non-statistical differences between the IBS and nonIBS datasets. The analysis is restricted to the proteins without hydrophobic protrusion at the IBS. The composition in secondary structures is reported in Supporting Information (S3B and S3C Fig).

odds ratio is positive showing, as for proteins with hydrophobic protrusions (Figs 4B and 5) that they are more likely to be found at the IBS. This is not the case for non-aromatic hydrophobes like Ile and Leu.

## 5. Analysis per superfamily

**5.1. Extended dataset with AlphaFold models.** Here we investigate the IBS dataset in each of the superfamilies. Since the number of structures is low for certain superfamilies (Fig 6B) lowering the statistical significance of an analysis per superfamily, we expand our dataset using structural models extracted from the AlphaFold database [31] (cf. Material and Methods section). Nearly 90% of the residues have a pLDDT over 70, indicating that the corresponding region is confidently modeled, while 10% have plDDT below 70 (S4A Fig). The pLDDT score distribution is comparable between superfamiles except for models of the annexin superfamily which have very high pLDDT. Residues with a pLDDT score below 70 are mostly outliers and were excluded from further analyses.

We checked that the AlphaFold models display IBS properties similar to the experimental structures. For that we take advantage of the 219 domains present in both our original dataset and in the AlphaFold dataset. We compare for those proteins the number of protrusions detected on the models and on the corresponding experimental structure. We perform a pairwise two-sided Wilcoxon signed rank test for each superfamily. The P-values reported in Table 1 show no significant difference between the two sets for PH, C2, START, C2DIS, PX, and annexins, so we included those AlphaFold models in our extended datasets. C1 models were left out of the dataset as the P-value was low (0.03). Models in the PLA and PLD families were not included either as there are no structures common to the two datasets. The final number of AF2 structural models and total number of structures per superfamilies are given in Supporting Material (S1 Table, S5 Fig). The number of structures for PH, C2 and PX are multiplied by 3 to 4 while the increase in other superfamilies is more moderate with 10–20% more structures. Interestingly, when we include Alphafold models, the number of structures without hydrophobic protrusions at the IBS drops from 39% to 30% (20% for the AlphaFold models only), still within the range of 1/3 of PMPs which we found in our earlier work [1].

**Table 1. P-values of Wilcoxon signed rank test between the pairwise samples of the number of hydrophobic protrusions between CATH structures (S100) and AlphaFold models.** A P-value over 0.05 indicates no statistical difference between the two samples. The last column indicates the number of structures in common between CATH (S100) and Alphafold subsets.

| Superfamily | P-value | Number of structures in common |
|---|---|---|
| ANNEXIN | 0.12 | 11 |
| C1 | 0.03* | 19 |
| C2 | 0.89 | 47 |
| DISC2 | 0.42 | 11 |
| PH | 0.69 | 97 |
| PLA | No overlap | 0 |
| PLD | No overlap | 0 |
| PX | 0.41 | 25 |
| START | 0.25 | 9 |

**5.2. Analysis per superfamily.** We calculated the distribution of amino acid types (Fig 8) in each superfamily using the extended dataset, i.e including the AlphaFold models. While we observe differences between superfamilies (see below), there are clear general trends. All domains except for ANNEXIN and C2 have more exposed basic than acidic amino acids. Among basic amino acids there are always more lysines than arginines (Fig 8B). There isn't such a clear trend across superfamilies for acidic amino acids Asp and Glu. Among hydrophobic amino acids, the ratio of aromatics is close to the ratio of non-aromatics (Leu, Ile, Cys, Met) but the variations can be large from one superfamily to another. Leu is more common than Ile.

The nine superfamilies have different levels of hydrophobic protrusions in their IBS (Cf. Figs 6 and S5), indicating that hydrophobic contribution to their membrane mechanisms vary between superfamilies.

Only ca. 60% of the annexin domains in the extended dataset have hydrophobic protrusions (S5 Fig), and they also are the superfamily with the lowest ratio of exposed hydrophobes at

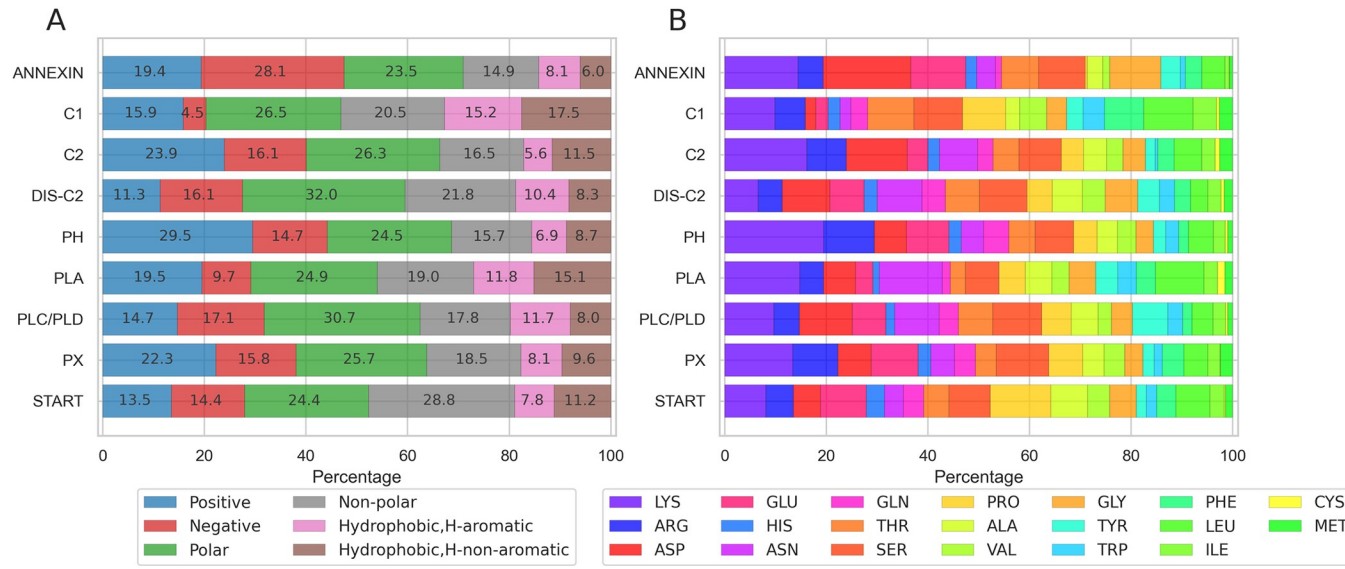

**Fig 8. Analysis per superfamily of the exposed residues at the IBS.** Composition of the exposed IBS stacked per superfamily and per amino acid type (A) and name (B) colored according to the "shapely" rastop color scheme [35].

their IBS (Fig 8). Annexin units are large (ca. 300 amino acids) and tend to bind specifically to acidic phospholipids. Their binding to membranes depends on calcium ions involved in membrane-binding [36–38]; five $Ca^{2+}$ ions are typically chelated by acidic amino acids (Asp, Glu). This is likely to be the explanation why acidic amino acids constitute a large share (28.1%) of all exposed amino acids at the IBS in the superfamily (Fig 8). The share of exposed Asp and Glu in the Annexin superfamily is much larger than for all other superfamilies, that have at most 17% of their exposed IBS residues being acidic amino acids. C2 domains also bind to membranes in a $Ca^{2+}$-dependent manner, often to phosphatidylserine lipids, and we observe a high prevalence of acidic amino acids at their IBS but lower (16.1%) than that of annexins. This could be reflecting the fact that C2 domains bind only two calcium ions or that there may be different membrane-binding mechanisms within the superfamily [39]. It also suggests that C2 domains might bind membranes with a higher hydrophobic contribution and a deeper insertion in the membrane. Indeed, their surface contains more exposed hydrophobes and over 80% of the domains in the extended dataset have one or more hydrophobic protrusions.

PH domains are the superfamily with the highest ratio of exposed basic amino acids (Lys, Arg) at their IBS (29.5%, Fig 8). Moreover, 70% of the domains in the extended dataset (80% in experimental structures) have one or more hydrophobic protrusions and the neighborhood of these protrusions is also rich in Lys and Arg (S5 Fig). This observation reflects the presence of the phosphatidylinositolphosphate (PIP) binding motifs consisting of lysines and arginines [40]. PX domains also display a high ratio of Lys and Arg at their IBS (22.3%) (Fig 8), but lower than for PH domains. Like PH domains, PX are PIP-binding domains [41, 42] but they are thought to not rely primarily on affinity with the lipid headgroup, but also take advantage of other forces including unspecific electrostatics and membrane insertion [42, 43]. We indeed observe that 70% of PH domains have hydrophobic protrusions (Figs 6 and S5). The neighborhood of those protrusions also has a high prevalence of Lys and Arg (S6 Fig), indicating that PIP recognition and unspecific electrostatics are also important forces in their membrane binding mechanism. It is also important to note that PH and PX have a higher prevalence of exposed Arg in their IBS than all the other superfamilies, in agreement with the presence of Arg in the PIP-binding sites.

C1 domains have the largest ratio of exposed hydrophobic amino acids at their IBS (17.5%) and the highest ratio of aromatics (15.2%). C1s bind specifically diacylglycerol (DAG) or phorbolesters to a binding site situated between two loops forming the IBS of C1 domains. The DAG-binding site is surrounded by a ring of exposed hydrophobic amino acids thought to penetrate the membrane [3, 44], in line with our observation of a high prevalence of hydrophobes. C1 domains are also described as binding specifically to phosphatidylserine (PS) containing membranes but our data does not show a high prevalence of Lys and Arg, which represent only 15.9% of the exposed IBS residues. This number is even lower (11.3%) for Discoidin C2 domains, which are also PS-specific [45]. The low ratio of basic amino acids could reflect the fact that the PS binding sites might contain other residues than Lys and Arg (eg. aromatics and Gln[45]), or that amino acids in the PS binding sites are not all solvent-exposed.

PLC/D, PLA and START are not membrane-targeting domains and their lipid-specificity varies vastly within their respective superfamily [46–48]. Moreover, the three domains might be associated with membrane-targeting domains such as C2 and PX/PH domains for PLD, or other domains for PLA, PLC and START. PLA and PLC/D are phospholipases, i.e enzymes, with a large variation in their substrate specificities. START domains are lipid transfer domains with diverse cargo selectivity and the organelle membranes they target vary within the superfamilies [46, 47]. For all these reasons, relating the observed properties of the IBS of each of these superfamilies to their respective function is challenging. However, we observe a few trends. Almost all PLA structures in our dataset have one or more hydrophobic

protrusions (Fig 6), and the exposed IBS is rich in exposed hydrophobes which represent nearly 27% of all exposed IBS residues, more than all other domains in our dataset, except for C1. The IBS of PLA domains are also rich in Lys and Arg, indicating a binding mechanism relying heavily on both hydrophobic insertion and unspecific electrostatics. The exposed IBS of PLD and START have very similar and balanced amino acid compositions, and they are among the superfamilies which have least structures presenting hydrophobic protrusions, with less than 45% in the dataset of experimental structures for START and 62% for PLDs (Fig 6).

## Discussion

We collected 1328 domain structures belonging to 9 superfamilies of PMPs including 6 membrane-targeting domains (Annexin, C1, C2, discoidin C2, PH, PX), 2 superfamilies of enzymes (PLA, PLC/D) and a superfamily of lipid-transfer proteins (START). Using information from the literature we annotated the IBS of representative structures in each superfamily (mainly in terms of secondary structure elements involved), and transferred that annotation to other domains in the same superfamily taking advantage of the structure alignment. Altogether the dataset of IBS amino acids counts 27012 entries, and the dataset of nonIBS amino acids, which we used as a reference in several analyses, counts 156998 amino acids. Further we extended the dataset adding model structures from 1194 domains extracted from the AlphaFold database, yielding 51689 amino acids in the IBS dataset and 271071 in the nonIBS dataset.

We analysed the dataset using our model of *hydrophobic protrusions*[1]. The rationale behind this model is that PMPs membrane binding sites often contain "hydrophobic spikes" [28, 29] or "protruding loops" [30]. *Protrusions* are thus a proxy to identify hydrophobic side chains at the tip of those protruding loops or amphipatic helices. In addition we analysed the *neighborhood* of protrusions and extended the concept of *hydrophobic protrusions* to any protrusion, thus capturing any amino acid type protruding at the IBS.

We confirm our earlier observation, namely that hydrophobic protrusions are over-represented at the IBS of peripheral membrane proteins, but we find IBS hydrophobic protrusions in less than two thirds of the domains in our datasets. This ratio varies between superfamilies from ca. 94% for PLA2 to only 50% of the START and Discoidin C2, and domains with IBS hydrophobic protrusions also have non-hydrophobic protrusions at their IBS (Fig 2). Those observations, and the variations in amino acid types in the environment of protrusions reflect the fact that PMPs bind to membranes with various mechanisms, using a combination of hydrophobic effect and nonspecific electrostatics [49], the respective importance of which may vary largely between proteins [22]. It is unclear yet how the balance between hydrophobic effect and electrostatics relate to the affinity of the domains for membranes but it was suggested–albeit on a very small dataset–that proteins binding with predominantly non-specific electrostatic might have lower experimental affinities than those binding with predominantly hydrophobic mechanism [49]. In addition, other more specific interactions may contribute to the overall affinity for membranes, such as the recognition of specific lipid headgroups by well-defined binding sites or calcium-dependent binding.

In proteins having at least one IBS hydrophobic protrusion, those are expectedly mostly located on loops and they consist of Leu, Ile, Phe, Tyr, Trp, Met. They are more likely to be found at the IBS than on the rest of the surface. Trp especially, with also Phe and Met, are characteristic of the neighborhood of IBS hydrophobic protrusions (Fig 5), and the three aromatics are characteristic of the neighborhood of non-hydrophobic protrusions (Fig 7). The most abundant Leu and Phe have sidechains likely to insert past the phosphate plane of lipid bilayers, while Trp and Tyr are usually found in the headgroup region and are important anchors for peripheral proteins [26, 27, 50–52]. In general, the neighborhood of hydrophobic

protrusions contains a mix of all amino acids types but a few patterns appear and in addition to the aromatics mentioned above we observe significantly more glycine in the IBS datasets than in the nonIBS dataset. Glycines in loops are likely to increase their flexibility [53, 54], a property that might be advantageous for protruding loops needing to adjust their conformation upon membrane binding. It might also be that their small size decreases steric hindrance and favors the presence of large hydrophobes on the protein surface.

Electrostatic interactions, and in particular basic amino acids such as Lys and Arg, have been described as playing an essential role for binding of PMPs to biological membranes [16, 17, 22, 49]. Interestingly, even in proteins having hydrophobic protrusions at their IBS, the most common amino acid on protrusions is Lys (Fig 4B). Protruding Lys are more likely at the IBS of PMPs than on the rest of their surface, just like hydrophobic protrusions are. In the experimental domain structures that have no hydrophobic protrusions at their IBS (39% of the dataset), Lys are commonly found protruding too. Arg, unlike Lys, are not more likely to be found at the IBS of PMPs and are generally less present at the IBS than on the rest of the surface. This is somewhat unexpected as both amino acids are positively charged. There is no large difference between the stability of Lys and Arg sidechains at membrane interfaces, neither from the Wimley White hydrophobicity scale for proteins at membrane interfaces [50] nor from predictions of their free energy of transfer from water to the membrane interface based on molecular simulations [51]. Actually, the guanidium group of Arg is thought to bind tighter to membrane interfaces than the ammonium group from Lys [55, 56]; guanidinium is more solvated than ammonium, it can form multiple hydrogen bonds with the lipid headgroups and lead to more membrane deformation than ammonium, explaining the importance of Arg -and not Lys- for the activity of cell penetrating peptides or antimicrobial peptides [57–60]. We pose the hypothesis that Lys is more favorable than Arg for the peripheral proteins in our dataset, who have evolved to bind to membranes but not to deform them or create pores as membrane-active peptides have.

A closer look at the prevalence of hydrophobic protrusions and amino acid composition in each superfamily offers a glimpse of how membrane-binding mechanisms vary between superfamilies. For example we find a large ratio of negatively charged amino acids (Asp and Glu) in annexin and C2 superfamilies whose membrane-binding mechanism is calcium-dependent [2, 3, 61, 62]. Asp and Glu chelate calcium ions mediating interactions with the membrane lipids.

Our observations within superfamilies are in good agreement with the known function of the domains, indicating that our strategy captures the IBS correctly. It is important because the strategy we used to build the datasets is relying on a number of approximations necessary to gather a large set of PMP structures.

One approximation is that the IBS for each domain is defined as a region on the protein surface instead of a collection of experimentally identified hotspots [63] because (1) such data is rare (available for about 50 proteins [63]) and (2) studies identifying hotspots are not systematic alanine-scanning so they are unlikely to map all amino acids involved in membrane-binding. Our strategy is justified by the fact that the protein-membrane interface is not restricted to a few hot-spots amino acids as in protein-ligand interfaces. Instead, the membrane is a surface which might get in contact with a fairly large patch of amino acids of a PMP, also beyond those that inserts in the membrane, and our study aims at characterizing the whole region.

Another approximation is that the study is performed on X-ray structures of the protein in their soluble state as this is the only state we have structural data for. Some PMPs might undergo structural changes upon membrane binding and, as a consequence, buried amino acids could become exposed. We verified that including the buried amino acids in the neighborhood of protrusions did not modify the outcome of the analysis. This indicates that

hydrophobes are mostly already exposed in the soluble forms of PMPs, which makes sense to take advantage of the hydrophobic effect.

Altogether the analyses of our datasets reveal characteristics of IBS that are in excellent agreement with what we know of PMP membrane binding mechanisms and functions. As such our results support the approach used to collect the IBS PMP datasets which are a useful resource that can be developed further, given the lack of experimental structural data on PMP-membrane interfaces.

## Conclusion

We collected a dataset of 1328 experimental structures and 1194 AphaFold models of peripheral proteins belonging to 9 distinct superfamilies. We established a computational strategy to identify and define their IBS, given the sparsity of experimental data on PMPs. The dataset was analyzed using our model for hydrophobic protrusions [1] which we extended to also include neighboring amino acids. Our observations are in good agreement with existing knowledge of PMPs IBS thus validating our approach: only 2/3 of the proteins in the dataset have protruding hydrophobes at their IBS, aromatic amino acids are more likely to be found in the IBS than on the rest of the proteins' surface and basic amino acids appear as a strong signature of PMPs IBS. Our analysis also reveals a number of novel patterns. Lysines are more likely to be found at IBS than arginines, unlike the predominance of Arg in membrane-active peptides. Glycine are over-represented in PMPs IBS possibly procuring IBS loops a much needed flexibility to insert in-between membrane lipids. Finally, in our analyses of the individual superfamilies, we observe amino acid distribution patterns in agreement with what is known about their membrane binding sites and function. Our dataset and approach thus hold great promises to investigate PMPs IBS, or for developing PMPs prediction tools using for example, machine learning approaches [63].

## Material and methods

### 1. Data collection and processing

We selected 9 protein superfamilies using the following criteria. Their IBSs are known, structurally conserved within protein superfamilies and there exists sufficient structural data to derive statistics from the dataset. Moreover these superfamilies should represent the diversity of PMP functions. PH, PKCa-C2, Factor V Discoidin C2, PKCd-C1, PX are membrane targeting domains. Phospholipases C/D and phospholipases A are enzymes. START domains are lipid transfer proteins. Annexins may act as scaffolding proteins or membrane organization.

*Experimentally determined structures.* Domain structures were collected from the CATH database (version 4.2.0) [64, 65] and the corresponding accession numbers are reported in Table 2. The PH superfamily in the CATH database contains protein domains others than actual pleckstrin-homology membrane targeting domains; PTB, RanDB, EVH/WH1 indeed share the same architecture than PH domains. We removed every PDB associated with their PFAM ID (PTB: PF00640; RanDB: PF00638; EVH/WH1: PF00568), as well as every structure belonging to the same cluster at 60% sequence similarity (S60). In total, 142 structures were removed from the 2.30.29.30 superfamily. The list of domains is provided as Supporting Information (S1 File).

Each structure was preprocessed to remove any alternative atom positions (only positions noted A are kept). Terminus oxygen atoms were renamed OXT to avoid issues with secondary structure calculations. We used Python 3.7.9 as programming language for data manipulation and analysis. Each structure was represented as a Pandas (1.1.5) [66] dataframe where each column is a PDB field and each row an atom. PDBs were converted using BioPandas 0.2.7 [67] and concatenated in a single dataframe.

**Table 2. Protein superfamilies included in the dataset with their accession numbers, representative structure and reference amino acids used to define the IBS.** Interpro SUPERFAMILY IDs in bold indicate that we removed long unstructured linkers from some of the AlphaFold models in that superfamily.

| Peripheral protein | CATH superfamily ID | Representative structure ID | Reference amino acids | PROSITE ID | Interpro SUPERFAMILYID |
|---|---|---|---|---|---|
| PH | 2.30.29.30 | 2da0A00 | K19; S42; P50 | PS50003 | **SSF50729** |
| PKC-C2 | 2.60.40.150 | 1rsyA00 | P169; D178; H237 | PS50004/ PS51547 | SSF49562 |
| C1 | 3.30.60.20 | 1ptrA00 | N237; F243; Q257 | PS50081 | - |
| PX | 3.30.1520.10 | 1h6hA00 | R33; G74; E100 | PS50195 | **SSF64268** |
| START | 3.30.530.20 | 2e3mA00 | G412; T448; N515 | PS50848 | **SSF55961** |
| DISCOIDIN-C2 | 2.60.120.260 | 1czsA00 | K23; N45; C76 | PS50022 | SSF49785 |
| Phospholipase C/D | 3.20.20.190 | 3rlhA00 | K59; I198; G205 | PS50035 | SSF51695 |
| Phospholipase A | 1.20.90.10 | 1pocA00 | L7; D76; D92 | PS00118 | **G3DSA:1.20.90.10** |
| ANNEXIN | 1.10.220.10 | 1a8aA01 | A25; K68; K77 | PS00223 | SSF47874 |

*Alphafold models*. We used PROSITE [68, 69] to identify sequence IDs of protein models to be retrieved from the AlphaFold Protein Structure Database [31]. In order to extract only the relevant structural domain from each AlphaFold model, we used the PROSITE annotations contained in the multiple sequence alignment headers. We also removed long unstructured linkers within domains using the Superfamily annotation from InterPro [70]. We used the PROSITE annotations for start and end positions of the domain structure in the AlphaFold models. The pLDDT score was used to assess local models quality and residues with pLDDT score below 70 were removed from the dataset (cf. S4 Fig)

## 2. Interfacial Binding Site selection, IBS and nonIBS datasets

Identification of the IBS in each family was based on the literature [3, 26, 27, 71–73], and on visual inspection of the representative structures (listed in Table 2) and their superimposition with other domains in the same superfamily. The same protocol was used for the experimentally-determined structures and the AlphaFold models.

We used the literature to identify the secondary structure elements (SSE) (eg loops or short amphipatic helices) involved in membrane binding for each representative domain (cf. Fig 1 and Table 2). This approach was chosen because experimental data on IBS individual amino acids is sparse and likely to be too restrictive [1]; data is available for only a small subset of PMPs and when available, it is only reported for a few amino acids while in reality the region in contact with the membrane is large. SSEs are also easy to identify in a fold and therefore provide a definition of the IBS particularly suitable for analysis within superfamilies of proteins. In the absence of sufficient sequence conservation in loop regions, we rely on structure similarity to transfer the IBS annotation between superfamily members.

In practice we define a plane to delimit in space the SSEs identified as forming the IBS, and the rest of the protein; the plane is defined by three amino acids positioned on the relevant SSEs (*cf. reference amino acids* in Table 2). We defined this plane for one representative protein in each superfamily (cf. *representative structure ID* in Table 2). All structures within a superfamily were then pairwise aligned with cath.superpose and the sequential structure alignment program for protein structure comparison (SSAP) [74]. The plane defined on the representative domain was used to delimit the IBS and non-IBS regions for each domain in the same superfamily (Fig 1). Gathering this annotation for all domains in the datasets lead to two sets of amino acids coined the IBS and nonIBS datasets. We provide these datasets as Supporting Information (S2 File).

## 3. Computation of structural features

**3.1. Secondary structures and Solvent Accessible Surface Area.**   Secondary structures were assigned with DSSP 3.0.0 [75, 76] and sorted into 3 categories: Helix (alpha-helix, 3/10 helix, 5-helix); β-sheet (β-bridge residue, extended strand) and loops (bend; turn and coil). The Accessible Surface Area (ASA) in Å$^2$ per residue for the main chain and the side chain were calculated with FreeSasa 2.1.0 [77]. The total ASA value was also added to the dataset. The relative accessible surface area RSA($r$) for an amino acid $r$ was computed from the total ASA of residue $r$ and its maximum theoretical SASA value in a tripeptide GLY-$r$-GLY as defined by Tien *et al.* [78]. A residue $r$ is considered solvent-exposed if RSA($r$) is over 20%.

**3.2. Protrusions, hydrophobic protrusions and their neighborhood.**   For every protein, alpha-carbons (C$_\alpha$) and beta-carbons (C$_\beta$) have been retained to compute a convex hull from their cartesian coordinates with the Sci-py QHull implementation [79, 80]. Based on our hydrophobic protrusion model [1], a protrusion (or protruding residue) is an amino acid whose C$_\beta$ atom is a vertex of the convex hull and counts less that 25 C$_\alpha$ or C$_\beta$ atoms within 1 nm of its C$_\beta$ atom. The hydrophobic character of protrusions is based on the Wimley and White hydrophobicity scale for proteins at membrane interfaces [50] (*cf.* S3 Table). Fig 1 shows a representation of the convex hull and hydrophobic protrusions for the reference structure of every superfamily. The neighborhood of a protrusion is defined as the ensemble of solvent exposed amino acids (RSA > 20%) whose C$_\alpha$ or C$_\beta$ are within 1 nm of the C$_\beta$ of the protruding amino acid. Naturally the protrusions themselves are excluded from the computed neighborhood.

**3.3 Redundancy.**   To remove the bias induced by sequence and structure redundancy, we used the CATH classification (SOLID), which uses a multi-linkage clustering based on similarities in sequence identity [81]. Clusters at 35%, 60%, 95% and 100% sequence identity are available. Our analyses are performed using one structure per S100 cluster. For reproducibility purposes and consistency with the CATH database version, the chosen structure is the reference structure (Cf Table 2) or alternatively, the first one in alphanumerical order.

To avoid bias from highly populated groups of domains in the PDB, it would have been preferable to use clusters with lower sequence identity but would yield too small datasets for this study. Yet, the analysis of the AlphaFold models showed similar patterns in structural and amino acid distributions than the patterns we observed for the PDB structures only. We are therefore confident that our results are not significantly biased towards proteins that are most abundant in the PDB.

## 4. Statistical analysis

*Frequency of hydrophobes among protrusions or solvent-exposed residues*. We measured the frequency of hydrophobic amino acids that are also solvent-exposed (noted hydrophobe ∩ exposed) or on protrusions (hydrophobe ∩ protrusion) relative to their location (IBS or not). The frequency $\hat{f}_{h \cap c}$ of residues being hydrophobic with respect to a reference property $C$ in our dataset is calculated as:

$$\hat{f}_{h \cap c} = \sum \frac{|G_h \cap G_c|}{|G_c|} \tag{1}$$

Where $G$ is a protein, $G_h$ is the set of residues on protein G that are hydrophobic, and $G_c$ the set of residues in protein $G$ meeting criteria $C$ (residue is a protrusion, or solvent-exposed).

*Odds ratio*. To estimate if amino acids with property B (eg. a particular amino acid type) are more likely to be present at the IBS (property A) or on the rest of the protein surface, we

calculated odds ratio (OR):

$$OR(A \cap B) = \frac{TC_{11} * TC_{00}}{TC_{10} * TC_{01}} \tag{2}$$

where TC is the contingency table, the first subscript indicates the localization (property A: 1 = IBS, 0 = non-IBS) and the second subscript is for property (B), such as being one of each of the 20 amino acid types (Fig 3B) or being located on a loop (Fig 4A).

As $OR \in \,]0; \infty]$ is centered around 1, we calculated and report log($OR$). Positive log($OR$ ($A \cap B$)) indicates that amino acids with property A are more likely to be present at the IBS.

The odds ratio and its two-sided P-values are computed from the contingency table with the "fisher_exact" module from the Python Scipy package and the 95% confidence interval (CI) is calculated as:

$$CI = e^{\log(OR) \pm 1.96 * SE} \tag{3}$$

With SE the standard deviation:

$$SE = \sqrt{\frac{1}{TC_{00}} + \frac{1}{TC_{01}} + \frac{1}{TC_{10}} + \frac{1}{TC_{11}}} \tag{4}$$

## 5. Implementation

All other analyses were implemented by us in a Python 3.7.9 package which was used in Jupyter lab notebooks [82], available on github (https://github.com/reuter-group/tubiana_etal_2022). Secondary structures were computed using DSSP 3.0 [75, 76] from Biopython 1.79 API [83]. Structural data were gathered from CATH 4.2.0 [64, 65]. PDBs were converted using Bio-Pandas 0.2.7 [67] into a DataFrame that was handled with Pandas 1.1.5 [66] and Numpy 1.19.4 [84]. Odds ratios and their associated P-values were calculated using the Scipy.stats.fisher_exact module [79]. Shapiro normality test [85] and Mann-Whitney statistical test [86] were computed with the Scipy.stats module. All graphics were generated with Matplotlib 3.3.3 [87] and its Seaborn interface (0.11.0) [88]. Images of protein structures were generated using Pymol [33] and our hydrophobic protrusion model viewer using Mol* [34] (https://reuter-group.github.io/peprmint/pepr2vis).

## Supporting information

**S1 Fig. Exposed amino acids in the IBS in the augmented dataset (CATH + Alphafold models) of IBS amino acids.** Composition is calculated across all superfamilies and grouped by (A,C) amino acid properties (positive, negative, polar, nonpolar) and (B,D) the 20 amino acids types for amino acids belonging to the exposed IBS surface (A,B) and exposed non IBS surface (C,D).
(TIF)

**S2 Fig. amino acid composition per secondary structure.**
(TIF)

**S3 Fig. Amino acid composition of protrusion and their neighbourhood.** Secondary structures composition of (A) the environment of hydrophobic protrusions in protein with hydrophobic protrusion at their IBS, (B) all protrusions and (C) their environment in proteins without hydrophobic protrusions at their IBS.
(TIF)

**S4 Fig. pLDTT of alphafold models in our dataset.** (A) Distribution of the pLDDT score per quality range. (B) Alphafold score per residue. The horizontal red dashed line represents the threshold at 70 for which the region may be unstructured in isolation according to Alphafold authors [31].
(TIF)

**S5 Fig.  Total number of structures in each of the superfamilies in the extended dataset** (A) CATH and Alphafold database and (B) Alphafold models only, with their respective percentage of structures without hydrophobic protrusions in the IBS (light gray).
(TIF)

**S6 Fig. Analysis per superfamily of the exposed neighbourhood of hydrophobic protrusions at the IBS.** Values are stacked per superfamily and per amino acid type (A) and name (B) colored according to the "shapely" rastop color scheme.
(TIF)

**S1 File. List of CATH domains and AlphaFold models used in our study.** Data available at https://github.com/reuter-group/tubiana_etal_2022/blob/main/Ressources/datasets/S1%20File.csv.
(CSV)

**S2 File. Dataset of amino acids of all included domains.** The file contains all amino acids in the IBS and nonIBS datasets, each annotated with structural features. Data available at https://github.com/reuter-group/tubiana_etal_2022/blob/main/Ressources/datasets/S2%20File.csv.zip.
(CSV)

**S1 Table. Origin of protein sequences (taxon levels 0 and 1) and number of structures.**
(XLSX)

**S2 Table. Dataset feature list and description.**
(XLSX)

**S3 Table. Amino acids classification.**
(XLSX)

## Acknowledgments

The authors thank Edvin Fuglebakk, Kamilla Jansen and Dandan Xue for fruitful discussions.

## Author Contributions

**Conceptualization:** Thibault Tubiana, Christine Orengo, Nathalie Reuter.

**Data curation:** Thibault Tubiana, Nathalie Reuter.

**Formal analysis:** Thibault Tubiana.

**Funding acquisition:** Nathalie Reuter.

**Investigation:** Thibault Tubiana.

**Methodology:** Thibault Tubiana, Ian Sillitoe, Christine Orengo, Nathalie Reuter.

**Project administration:** Nathalie Reuter.

**Resources:** Thibault Tubiana, Ian Sillitoe, Christine Orengo, Nathalie Reuter.

**Software:** Thibault Tubiana.

**Supervision:** Nathalie Reuter.

**Validation:** Thibault Tubiana.

**Visualization:** Thibault Tubiana, Nathalie Reuter.

**Writing – original draft:** Thibault Tubiana.

**Writing – review & editing:** Thibault Tubiana, Ian Sillitoe, Christine Orengo, Nathalie Reuter.

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
