## [Decision Letter · Decision Letter 0]

6 Sep 2022

Dear Prof. Reuter,

Thank you very much for submitting your manuscript "Dissecting peripheral protein-membrane interfaces" for consideration at PLOS Computational Biology.

As with all papers reviewed by the journal, your manuscript was reviewed by members of the editorial board and by several independent reviewers. In light of the reviews (below this email), we would like to invite the resubmission of a significantly-revised version that takes into account the reviewers' comments. In the revised version please emphasize the novelty of the current results relative to current knowledge of PMPs.

We cannot make any decision about publication until we have seen the revised manuscript and your response to the reviewers' comments. Your revised manuscript is also likely to be sent to reviewers for further evaluation.

Sincerely,

Dina Schneidman

Software Editor

PLOS Computational Biology

Reviewer's Responses to Questions

**Comments to the Authors:**

Reviewer #1: In this study, Tubiana et al build on their previous published work [Fuglebakk et al 2018], in the same journal. They characterise the basis of interfacial binding sites (IBS) for a range of peripheral membrane proteins (PMPs), using an impressive number of proteins for their analysis. In particular, the study is able to take advantage of the recent release of a large number of AlphaFold (AF) structures, so is very timely.

In general, the study is well conducted and rigorous, with some interesting insights. I recommend publishing with a few small changes:

Larger comments:

Firstly, I’m a little unclear on exactly how the IBS are determined. It’s stated that the structural database includes known IBS, but seeing as this is an integral part of the analysis here, it would be good if this were expanded. If, for instance, the structural IBS were predicted using bioinformatics, is there a risk that the analyses here are biased by previous assumptions? Similarly, how were the IBS for the AF models determined?

The analysis is done on completely rigid structures, solved in most cases I presume in a soluble state. I feel that this represents a possible limitation: if the IBS changes upon membrane binding (especially seeing as many sites contain loops) might this skew or change the data at all? For instance, could hydrophobic residues become more prominent upon binding? It would be good if this was addressed in the discussion.

Whilst the AF models are a really nice addition to the story, is there a risk that the relatively low accuracy of the AF side chains might skew an analysis of “protrusions”? Could this explain the difference between the AF and non-AF structures in terms of hydrophobic protrusions at the IBS (20 vs 39%)?

As an additional thought (not necessary for publication, but interesting) it is known that some domains bind to membrane with far higher affinities than others. Can this knowledge feed into the analysis here? i.e. do the proteins with no hydrophobic protrusions at the IBS fall into families which typically have lower binding affinities? Or do PMPs with more Arg have higher affinities, which is why many PMPs have fewer Arg (so they don’t bind too tightly?)

Smaller comments:

• The number of AF models used is reported to be 1423 in the intro, but 1194 in the abstract and discussion. Is this a discrepancy?

• The figures were a little blurry and the graph labels were often difficult to read. I recommend larger labels and higher resolution for the final submission

• The provided GitHub link doesn’t work (pepr2ds). I think I found it, but this should be changed in the manuscript.

• P5 line 108: “we extent this dataset” > “we extend this dataset”

Reviewer #2: The work is technically well done and provides important data and models on amino acids and regions of proteins and domains that interact with lipid membranes.

The main shortcoming of the work is the novelty and significance. The peripheral proteins and domains that bind lipid membranes have been studied in great detail both computationally and experimentally. The work presented herein is technically solid and important but doesn't make a significant advance in new information on lipid binding amino acids or composition of the interfacial binding surface amino acids. Hydrophobic and aromatic amino acids have been identified extensively in previous studies. This is good as the computational work and models confirm what has previously been measured/known but doesn't advance the field in significantly new ways.

Reviewer #3: This article presents the collection of a dataset of peripheral membrane proteins (PMPs) belonging to 9 superfamilies. For each PMP, the authors characterized the interfacial binding site (IBS) and split the dataset into two datasets: the amino acids in the IBS and the rest amino acids. The authors analyzed the amino acid distribution in these datasets and applied a mathematical model of hydrophobic protrusions that the authors present in “Fuglebakk et al. PLoS Comput Biol 2018”. New patterns were discovered in the IBS, such as the over-representation of glycine amino acids and the preference for lysine over arginine amino acids. I believe that this new PMP dataset will contribute to the creation of novel machine learning algorithms and PMP prediction tools. I would therefore recommend the publication of the manuscript. However, some major comments must be addressed to improve the quality of the manuscript:

1) The manuscript is about “Dissecting peripheral protein-membrane interfaces” but as it is also mentioned by the authors “the IBS for each domain is broadly defined and is likely to include more amino acids than the ones actually interacting with the lipids”. I understand that it is not possible to manually dissect ~2500 proteins but since other researchers might use this dataset for training machine learning algorithms or validating prediction models, more emphasis should be given to removing from the IBS the amino acids that do not contribute to the protein-membrane affinity.

2) Most of the analysis is performed on the mathematical model of hydrophobic protrusions that the authors present in “Fuglebakk et al. PLoS Comput Biol 2018” instead of performing analysis on the dataset itself. It is like the purpose of the manuscript has changed to test the model of hydrophobic protrusions in a new dataset. The only analysis performed in the dataset is the amino acid distribution in the IBS vs the nonIBS sets. More analysis between all or the solvent-exposed amino acids in the IBS vs the nonIBS would provide better insights into important properties of the IBS. For example, the secondary structure elements between the IBS and nonIBS, the amino acid conservation between the IBS and nonIBS in each superfamily, etc.

I have also some minor comments and suggestions to improve the clarity of the presentation:

1) Main text, Page 11, Lines 237-243: I cannot understand why the fact that “proteins without hydrophobic protrusions at their IBS tend to have less protrusions in their IBS in general” is significant. The authors should elaborate more on the scope of this analysis.

2) The fact that glycine amino acids are over-represented in the IBS is interesting. Indeed, one possible explanation is that glycine amino acids provide flexibility, which helps hydrophobic loops to insert the membrane. Do you think that the absence of a side chain in glycine amino acids drives nearby hydrophobic and aromatic amino acids to be more solvent-exposed and thus it is easier for them to anchor on the membrane? If yes, you could also discuss it in the manuscript.

3) In Table 1 the column “Number of structures in common” should be described in the caption.

4) I suggest adding colors to Figures 5B, 7A, and S4.

5) In my opinion, the main text contains a lot of figures and tables. I suggest moving Figure 7 and Table 3 to the SI.

6) Main text, Page 25, Line 567. I believe there is a typo here and instead of Fig. 1, it should be Fig. 9.

7) Reference 9 and reference 11 are the same.

**Have the authors made all data and (if applicable) computational code underlying the findings in their manuscript fully available?**

Reviewer #1: Yes

Reviewer #2: Yes

Reviewer #3: Yes

PLOS authors have the option to publish the peer review history of their article (what does this mean?). If published, this will include your full peer review and any attached files.

Reviewer #1: **Yes: **Robin Corey

Reviewer #2: No

Reviewer #3: No
---

## [Decision Letter · Decision Letter 1]

24 Nov 2022

Dear Prof. Reuter,

We are pleased to inform you that your manuscript 'Dissecting peripheral protein-membrane interfaces' has been provisionally accepted for publication in PLOS Computational Biology.

Best regards,

Dina Schneidman

Software Editor

PLOS Computational Biology

Reviewer's Responses to Questions

**Comments to the Authors:**

Reviewer #1: The authors have done an excellent job in responding to my suggested changes, and I feel they have now soundly addressed all the points raised by myself and the other referees. It is my opinion that the manuscript is ready for publication.

Reviewer #2: The authors have more clearly addressed some of my concerns on novelty and significance.

Reviewer #3: The authors answered all my comments and revised their manuscript accordingly. Thus, I believe that the manuscript is ready for publication.

**Have the authors made all data and (if applicable) computational code underlying the findings in their manuscript fully available?**

Reviewer #1: Yes

Reviewer #2: Yes

Reviewer #3: Yes

PLOS authors have the option to publish the peer review history of their article (what does this mean?). If published, this will include your full peer review and any attached files.

Reviewer #1: **Yes: **Robin Corey

Reviewer #2: No

Reviewer #3: No

---

## [Editor Report · Acceptance letter]

9 Dec 2022

PCOMPBIOL-D-22-01007R1 

Dissecting peripheral protein-membrane interfaces

Dear Dr Reuter,

I am pleased to inform you that your manuscript has been formally accepted for publication in PLOS Computational Biology. Your manuscript is now with our production department and you will be notified of the publication date in due course.

With kind regards,

Zsofia Freund
